# Simultaneous Localization and Guidance of Two Underwater Hexapod Robots under Underwater Currents

**DOI:** 10.3390/s23063186

**Published:** 2023-03-16

**Authors:** Jonghoek Kim

**Affiliations:** Defense System Engineering Department, Sejong University, Seoul 05006, Republic of Korea; jonghoek@gmail.com

**Keywords:** underwater localization, dual-robot exploration, underwater exploration, sea currents estimation, hexapod robots, cooperative exploration, cooperative localization

## Abstract

This paper addresses the simultaneous localization and guidance of two underwater hexapod robots under sea currents. This paper considers an underwater environment where there are no landmarks or features which can assist a robot’s localization. This article uses two underwater hexapod robots that move together while using each other as landmarks in the environment. While one robot moves, another robot extends its legs into the seabed and acts as a static landmark. A moving robot measures the relative position of another static robot, in order to estimate its position while it moves. Due to underwater currents, a robot cannot maintain its desired course. Moreover, there may be obstacles, such as underwater nets, that a robot needs to avoid. We thus develop a guidance strategy for avoiding obstacles, while estimating the perturbation due to the sea currents. As far as we know, this paper is novel in tackling simultaneous localization and guidance of underwater hexapod robots in environments with various obstacles. MATLAB simulations demonstrate that the proposed methods are effective in harsh environments where the sea current magnitude can change irregularly.

## 1. Introduction

In recent years, various autonomous unmanned robots have been developed for military, environmental and robotic research applications. Consider a scenario where autonomous underwater robots are deployed to approach a predefined goal while avoiding collision with obstacles, such as underwater nets. This scenario is feasible for a large variety of marine tasks, such as underwater construction, oceanographic surveys, coastal patrol, seabed and bathymetric data collection [1,2].

The localization of an underwater robot in deep water is not trivial, since the signal of Global Positioning Systems (GPS) cannot reach a robot in deep water. Doppler Velocity Logs (DVL) and Inertial Measurement Units (IMU) can be used to localize an underwater vehicle. However, the localization error using these sensors increases gradually as the travel distance of the robot increases [3]. This error integration is inevitable since the position estimation can be derived by integrating velocity measurements of DVL or integrating acceleration measurement of IMU twice. Moreover, DVL cannot be used for localization of a robot that crawls on the sea bottom.

Simultaneous localization and mapping (SLAM) was widely used to build a map of an unknown environment while simultaneously keeping track of an agent’s location within it. In SLAM, a robot keeps storing its trajectory, odometry information sequence, and observation information sequence. Then, graph-based optimization is used to update both the robot pose and the feature map, based on the stored information. As a robot explores a large outdoor environment, stored data increases and the computational load of the optimization increases. Therefore, the overall computational load of SLAM increases, as the robot travels a longer distance. It is thus argued that SLAM may not be preferred as a robot with small embedded computers explores a large outdoor environment, such as a seabed.

Various Visual-Inertial SLAM (VI-SLAM) methods [4,5,6] have been applied for robot localization. The authors of [7] review various papers on VI-SLAM. VI-SLAM applied both a camera and an IMU for robot localization [4,5]. VI-SLAM methods require the calculation of the 6-DOF transformation between the camera and the IMU. However, calculating this transformation is computationally expensive [8]. Any SLAM method requires that a robot can extract features, such as corners of the obstacles, from the environment [6,9,10,11].

Unknown underwater environments, where unknown sea currents exist in the deep sea, are considered in this paper. A hexapod robot [12] is utilized, which can extend its leg into the sea bottom. Once a hexapod robot extends its leg into the sea bottom, it can stay still while overcoming underwater currents. The hexapod robot in [12] mimics the design of crabs and lobsters that live in stormy waters but are still able to control their movements. This paper handles the localization of an underwater hexapod robot with small embedded computers, whose mission is to explore a large seabed. A practical scenario is considered, where it is difficult for an underwater robot to derive stable features of the underwater environment. For instance, it is common for an underwater robot’s vision sensors to have low visibility due to dust in water. In addition, an underwater robot on a flat sea floor has difficulty finding a stable feature to assist the robot’s localization. It may not be feasible to detect a stable feature since underwater currents can change the shape of the sea floor. In the case where there are no stable features in the environments, it is not feasible to use SLAM algorithms. It is thus argued that SLAM is not feasible, as a hexapod robot explores the seabed where underwater currents can easily change the seabed shape.

It is acknowledged that an Acoustic Doppler Current Profiler (ADCP) can be used to measure the velocity of underwater currents in real-time. However, ADCP cannot be used to measure motion perturbation due to underwater currents. Even in the case where the velocity of underwater currents can be measured, it is not trivial to estimate the complex interaction between the underwater currents and the maneuver of a hexapod robot on the sea bottom. This paper uses the Ultra-Short BaseLine (USBL) sensors for simultaneous estimation of both a robot’s pose and the motion perturbation due to underwater currents. Note that ADCP measures the velocity of currents, but it cannot measure the robot’s motion perturbation due to underwater currents.

This paper considers an underwater environment where there are no stable features that can assist a robot’s localization. Moreover, due to underwater currents, dead reckoning localization based on inertial navigation sensors results in a severe accumulation of localization errors. In order to avoid localization error accumulation due to underwater currents, this article introduces using two hexapod robots. These two robots move together while using each other as static landmarks in the environment. This implies that two robots move as one pair while a moving robot uses another robot as a static landmark for localization.

A moving robot measures the relative position of another static robot, in order to estimate its position while it moves. Here, relative position (bearing and range) measurements are feasible considering an underwater robot equipped with USBL sensors. To the best of our knowledge, this paper is novel in using relative position measurements for the cooperative localization of underwater hexapod robots.

This article proposes a stop–go policy, which iterates the following process. The moving robot sets a new waypoint and moves to reach the waypoint. While the moving robot travels, it calculates the relative position of the anchor robot using USBL sensor measurements and localizes itself. After the moving robot reaches the waypoint, the moving robot extends its legs into the sea bottom and is set as a new anchor robot. Then, the previous anchor robot becomes a new moving robot. The new moving robot approaches the anchor robot while localizing itself. Once the moving robot reaches the anchor robot, we iterate the process in this paragraph. Iterate this until the robots visit the goal point.

Due to underwater currents, a robot cannot maintain its desired course. Sea currents may perturb the robot’s maneuver significantly. Moreover, there may be obstacles, such as underwater nets, that a robot needs to avoid. For instance, the robot can use active sonar sensors to detect the relative position of underwater nets [13]. This manuscript thus develops a guidance strategy for avoiding obstacles, while estimating the motion perturbation due to underwater currents.

In summary, the novelties of this article are summarized as follows:As far as we know, this paper is novel in addressing a stop–go policy for simultaneous localization and guidance of underwater hexapod robots.This paper is novel in using relative position measurements for cooperative localization of underwater hexapod robots.This article develops a guidance strategy for avoiding obstacles while estimating the motion perturbation due to underwater currents.

The performance of the proposed cooperative localization methods is verified using MATLAB simulations. MATLAB simulations verify that the proposed localization methods are effective in that the current magnitude changes irregularly.

The remaining of this paper is organized as follows. Section 2 presents the literature review on cooperative localization and introduces the preliminary information of this paper. Section 3 presents several definitions and assumptions. Section 4 presents the cooperative localization based on the stop–go policy. Section 5 introduces MATLAB simulation results to demonstrate the performance of the proposed motion control with cooperative localization. Section 6 provides the conclusions.

## 2. Literature Review on Cooperative Localization

In order to improve the localization of two hexapod robots, the coordinated motion of the two robots is utilized. Coordinating multiple robots can be utilized in many tasks, such as sensor network building [14], exploration [15], intruder capture [16,17], rendezvous [18,19], formation control [20,21], or target tracking [22,23].

References [24,25] presented cooperative localization (CL) of a robot assisted by another autonomous vehicle, say Communication Navigation Aid (CNA), which can use GPS to localize itself. The CNA serves as a leader, and the robot serves as a follower. Reference [25] discussed the positioning performance indicators of different positioning methods in the case of different numbers of outliers in the measurement information. References [26,27] considered the case where the movement of a follower is constrained by the maneuver of a leader. In [27,28], the CNA localizes a robot by measuring the relative position (or distance) of the robot. However, the CNA must exist near the sea surface in order to use GPS.

In Ref. [29], surface buoys and anchor nodes were used for the localization of underwater sensor networks. Surface buoys are equipped with GPS to obtain their location estimates. In addition, anchor nodes are powerful nodes that can make direct contact with the surface buoys, and are capable of self-localization based on such contacts.

Our article considers cooperative navigation in the case where there are no CNAs or surface buoys. This is feasible as robots operate in the deep sea. This article considers a scenario where it is not possible to establish a communication link between the CNA (or surface buoys) and a hexapod robot at the sea bottom.

Several papers handled cooperative localization of underwater robots based on bearing-only [30,31] or range-only measurements [32]. References [33,34,35,36,37] considered multi-robot systems, such that every robot has sonar sensors to measure the time of arrival (TOA) of sound emitted from a transmitter. By processing sonar measurements of every robot, the robot team estimates the transmitter’s location. The authors of [38] handled how to place multiple sensors so as to yield the best-expected source location estimate. Reference [1] considered the case where a robot uses mutual range measurements for cooperative localization. Reference [1] validated their cooperative navigation procedure using experiments. However, [1] considered localization of a torpedo-shaped robot with propellers, thus [1] cannot be applied for localization of a hexapod robot.

In the literature, many location strategies were utilized for underwater localization [39,40]. Short-baseline (SBL) [41] or long-baseline (LBL) [40,42] can be used for locating an underwater robot. However, SBL requires that an acoustic pulse is transmitted by the transceiver and detected by the subsea transponder, which replies with its own acoustic pulse. LBL and SBL require that additional structures are built for underwater robot localization. Therefore, LBL and SBL are not suitable for underwater robots which explore unknown environments.

As far as we know, our manuscript is novel in using relative position measurements for cooperative localization of underwater hexapod robots. Note that relative position measurements are viable using USBL sensors. Relative position measurements are more desirable than bearing-only or range-only measurements since the relative position information can be measured directly.

## 3. Definitions and Assumptions

This paper considers two hexapod robots that move on a flat sea bottom. This article considers the problem of making two robots reach the goal while localizing themselves in environments with obstacles.

We address notations related to vectors and matrices. Let v[j] define the *j*-th element in a column vector, say v. Let $\mathrm{diag}(a_{1},a_{2},\cdots ,a_{n})$ define a diagonal matrix where $a_{1},a_{2},\cdots ,a_{n}$ appear as diagonal elements in this order. Let $v_{1}\cdot v_{2}$ define the inner product between two vectors, say $v_{1}$ and $v_{2}$. Let $\tan^{-1}\left(y,x\right)$ denote the phase (or angle) of the complex number x+iy. Therefore, it is feasible that *y* or *x* in x+iy is zero. However, it is not possible that both *x* and *y* in x+iy are zeros.

Let $\hat{v}$ define the estimation of a variable v. Here, we use the hat operator to represent an estimation. The variance of $\hat{v}$ is defined as $E({\left(\hat{v}-E\left(\hat{v}\right)\right)}^{2})$. Here, +E(v) denotes the expected value (mean) of a random variable, say v. Let $V(\hat{v})$ define the variance of the estimate $\hat{v}$.

We consider two hexapod robots that move on a flat sea bottom. Hence, this paper considers the localization problem of a robot in 2D environments. We use an inertial reference frame whose origin is an arbitrary point on the flat sea bottom. The frame consists of two coordinates: one represents the position along the northern axis, and the other one is along the eastern axis. This frame is called the *North-East (NE) reference frame*.

In the NE reference frame, let $r_{i}\in R^{2}$, where i∈{1,2}, define the 2D position of the *i*-th robot. In other words, we need to derive ${\hat{r}}_{i}\in R^{2}$ while the *i*-th robot moves. Let $V({\hat{r}}_{i})$, where i∈{1,2}, define the variance of the estimate ${\hat{r}}_{i}$.

This paper considers discrete-time systems. Let *T* define the sampling interval in discrete-time systems. In the NE reference frame, $r_{i}\left(k\right)\in R^{2}$ defines the 2D position of the *i*-th robot at sampling-stamp *k*. In addition, let ${\hat{r}}_{i}\left(k\right)\in R^{2}$ denote the estimation of $r_{i}\left(k\right)$. Let $∥r_{i}\left(k\right)-{\hat{r}}_{i}\left(k\right)∥$ present the localization error of the *i*-th robot at sampling-step *k*.

### 3.1. Horizontal Motion Model of a Hexapod Robot

One considers the horizontal motion of a robot in the NE reference frame. In the case where there is no sea current, the motion model of the *i*-th robot is
(1)$$\begin{array}{c} r_{i}\left(k+1\right)=r_{i}\left(k\right)+T\times v_{i}\left(k\right)\mathrm{cs}\left(\theta_{i}\left(k\right)\right). \end{array}$$

Here, c(*)=cos(*), and s(*)=sin(*). In addition, $v_{i}\left(k\right)$ is the through-water speed of the *i*-th robot at sampling-stamp *k*, and $\theta_{i}\left(k\right)$ defines the orientation direction of the *i*-th robot at sampling-stamp *k*, measured counterclockwise from the East direction. Furthermore, $\mathrm{cs}\left(\theta_{i}\left(k\right)\right)=\left(\begin{array}{c} c(\theta_{i}\left(k\right))\\ s(\theta_{i}\left(k\right)) \end{array}\right)$ presents the orientation vector of the *i*-th robot. The simple dynamic model in (Equation 1) is commonly used in multi-robot systems [43,44,45,46,47,48,49].

It is acknowledged that the detailed inertial parameters of a hexapod robot are not considered in horizontal motion controls in (Equation 1). $v_{i}\left(k\right)\mathrm{cs}\left(\theta_{i}\left(k\right)\right)$ in (Equation 1) can be set as the input vector of the robot at sampling-stamp *k*.

Hexapod robots are omnidirectional vehicles [50]. Both sideways motion and rotational motion can be used to move according to the input vector [50]. References [50,51] discussed how to make the robot change foot placement based on an input vector in addition to controlling the position and orientation of the body.

Note that both $v_{i}\left(k\right)$ and $\theta_{i}\left(k\right)$ are not used in the localization process of underwater robots. These values are used to make a robot under underwater currents move towards its assigned waypoint.

In the NE reference frame, let $w\left(k\right)\in R^{2}$ denote the waypoint at sampling-stamp *k*. How to make the robot at sampling-stamp *k* move towards a waypoint w(k) is presented in Section 4.2. Since $v_{i}\left(k\right)$ and $\theta_{i}\left(k\right)$ are not used in the localization process, the error in measurements of $v_{i}\left(k\right)$ and $\theta_{i}\left(k\right)$ does not have an effect on the performance of the proposed localization scheme. The performance of the proposed localization scheme depends on the accuracy of sensors measuring the relative distance (bearing and range) between two robots.

In practice, there are sea currents in underwater environments. In the case where there are underwater currents, the motion model of the *i*-th robot is
(2)$$\begin{array}{c} r_{i}\left(k+1\right)=r_{i}\left(k\right)+T\times \left(f\left(r_{i}\left(k\right)\right)+v_{i}\left(k\right)\mathrm{cs}\left(\theta_{i}\left(k\right)\right)\right). \end{array}$$

This model is presented in [52]. In (Equation 2), $f\left(r_{i}\left(k\right)\right)\in R^{2}$ defines the robot’s horizontal motion perturbation due to underwater currents. Since a robot crawls on the sea bottom, it is not trivial to estimate $f(r_{i}\left(k\right))$ while the robot moves. How to estimate the horizontal motion perturbation $f(r_{i}\left(k\right))$ is discussed in Section 4.2.

See that the robot’s speed in (Equation 2) is $∥f\left(r_{i}\left(k\right)\right)+v_{i}\left(k\right)\mathrm{cs}\left(\theta_{i}\left(k\right)\right)∥$ which is different from it’s through-water speed $v_{i}\left(k\right)$. (Equation 2) shows that due to underwater currents, the orientation direction of the *i*-th robot does not coincide with the actual movement direction of the robot.

### 3.2. Relative Position Measurements between Two Robots

This paper uses the following stop–go policy. The proposed stop–go policy is as follows. During the maneuver of one moving robot, another robot stays still (extends legs into the sea bottom) and performs as the anchor robot. The moving robot struggles to move while being pushed by underwater currents.

During the maneuver of a moving robot, the anchor robot stands still. Assume that the anchor robot can stand still while extending its legs into the sea bottom. In this way, the anchor robot can resist severe underwater currents. The moving robot localizes itself by measuring the relative position of the anchor robot.

A moving robot uses USBL sensors for measuring the relative position (bearing and range) of the anchor robot. Let $r_{s}$ represent the maximum range of USBL sensors. Let $r_{c}$ represent the maximum communication range of a robot. Let $r_{cs}=min\left(r_{s},r_{c}\right)$. This implies that $r_{cs}$ is the smaller value between $r_{s}$ and $r_{c}$.

In the NE reference frame, let $r^{S}\left(k\right)\in R^{2}$ define the position of the anchor robot at sampling-stamp *k*. In the NE reference frame, let $r^{M}\left(k\right)\in R^{2}$ define the position of the moving robot at sampling-stamp *k*.

While the moving robot moves, it estimates its 2D position ${\hat{r}}^{M}\in R^{2}$ by measuring the relative position (bearing and range) of the anchor robot. Suppose that at each sampling-stamp *k*, the moving robot measures the bearing and range of the anchor robot.

The measurement model of a robot is as follows. Let $r^{rel}\left(k\right)=r^{S}\left(k\right)-r^{M}\left(k\right)$ for convenience. Moreover, let $b_{k}^{l}$ define the bearing measurement of the anchor robot with respect to the moving robot. The equation for $b_{k}^{l}$ is
(3)$$\begin{array}{c} b_{k}^{l}=\tan^{-1}\left(\left(r^{rel}\left(k\right)\right)\left[2\right],\left(r^{rel}\left(k\right)\right)\left[1\right]\right)+n_{k}. \end{array}$$

Here, $n_{k}$ has a Gaussian distribution with mean 0 and standard deviation brgS. In (Equation 3), recall that v[j] defines the *j*-th element in a column vector, say v.

The moving robot measures the range of the anchor robot. As the range measurement, we have
(4)$$\begin{array}{c} r_{k}^{l}=∥r^{rel}\left(k\right)∥+m_{k}. \end{array}$$

Here, $m_{k}$ has a Gaussian distribution with mean 0 and standard deviation rngS.

Let ${\hat{r}}^{S}\left(k\right)\in R^{2}$ define the 2D estimate of the anchor robot at sampling-stamp *k*. In the NE reference frame, the 2D position of the moving robot can be localized as
(5)$$\begin{array}{c} {\hat{r}}^{M}\left(k\right)={\hat{r}}^{S}\left(k\right)+\left(\begin{array}{c} r_{k}^{l}c\left(b_{k}^{l}\right)\\ r_{k}^{l}s\left(b_{k}^{l}\right) \end{array}\right). \end{array}$$

Using (Equation 5), one gets $V({\hat{r}}^{M}\left(k\right))$ as
(6)$$\begin{array}{c} V\left({\hat{r}}^{M}\left(k\right)\right)=V\left({\hat{r}}^{S}\left(k\right)\right)+V\left(\begin{array}{c} r_{k}^{l}c\left(b_{k}^{l}\right)\\ r_{k}^{l}s\left(b_{k}^{l}\right) \end{array}\right). \end{array}$$

Using partial derivatives of $r_{k}^{l}c\left(b_{k}^{l}\right)$ or $r_{k}^{l}s\left(b_{k}^{l}\right)$ with respect to $r_{k}^{l}$ and $b_{k}^{l}$, one derives
(7)$$\begin{array}{c} V\left(\begin{array}{c} r_{k}^{l}c\left(b_{k}^{l}\right)\\ r_{k}^{l}s\left(b_{k}^{l}\right) \end{array}\right)=M_{1,k}\mathrm{diag}\left(rngS^{2},brgS^{2}\right)M_{1,k}^{T}. \end{array}$$

Here,
(8)$$\begin{array}{c} M_{1,k}=\left(\begin{array}{cc} \frac{\partial (r_{k}^{l}c\left(b_{k}^{l}\right))}{\partial r_{k}^{l}} & \frac{\partial (r_{k}^{l}c\left(b_{k}^{l}\right))}{\partial b_{k}^{l}}\\ \frac{\partial (r_{k}^{l}s\left(b_{k}^{l}\right))}{\partial r_{k}^{l}} & \frac{\partial (r_{k}^{l}s\left(b_{k}^{l}\right))}{\partial b_{k}^{l}} \end{array}\right). \end{array}$$

### 3.3. Sinking Process of Two Hexapod Robots

Two hexapod robots are deployed from the sea surface. While the robots sink, their mutual distance may increase as time goes on, since the position of each robot is perturbed by underwater currents. It is desirable that the two robots are not too far from each other once they get to the sea bottom. Otherwise, two robots may not sense or communicate with each other after they get to the sea bottom.

We assume that each robot has depth sensors for measuring its depth. It is assumed that two robots are connected using a cable. The cable is used to avoid the case where the distance between the two robots gets too far while they fall to the sea bottom. In this way, the mutual distance between the two robots can be always shorter than the maximum length of the cable. It is assumed that the maximum length of the cable is shorter than $r_{cs}$. Recall that $r_{cs}$ is the smaller value between $r_{s}$ and $r_{c}$.

Initially, two robots settle down close to each other, using the cable. Since the maximum length of the cable is shorter than $r_{cs}$, the relative distance between the two robots is shorter than $r_{cs}$. Once two robots reach the sea bottom, the cable can be disconnected, since the cable is not used anymore. An entangled cable does not make problems, since the cable is disconnected after the robots reach the sea bottom.

While these robots fall to the sea bottom, the localization error of a robot increases due to underwater currents. This implies that a robot can be pushed by underwater currents while it falls to the sea bottom. However, its NE location estimate error can increase when it falls to the sea bottom.

At sampling-stamp 0, for two robots (${\hat{r}}_{i}\left(0\right)$ where i∈{1,2}), one initializes the variance of each robot as
(9)$$\begin{array}{c} V\left({\hat{r}}_{i}\left(0\right)\right)=P_{0}. \end{array}$$

Here, $P_{0}$ is a positive definite diagonal matrix and is set considering the increase of localization error while each robot sinks. The localization approach in [3] can be used, in order to locate each robot while it sinks. However, in this case, the localization error increases gradually as the depth of the robot increases. $P_{0}$ can be considered as the initial covariance matrix in Kalman filters [53].

For instance, one can assume that underwater currents can push a robot by at most 5 m, while the robot falls for every 100 m. This implies that as a robot falls for 100×2 m, underwater currents can push the robot by at most 5×2 m. MATLAB simulations consider the case where the robot falls for 200 m until reaching the sea bottom. Here, a robot’s depth change can be measured using its depth sensor. In this case, the robot can have been pushed by at most 10 m. Thus, MATLAB simulations set the initial covariance matrix as $P_{0}=\mathrm{diag}\left(10^{2},10^{2}\right)$.

### 3.4. The Summary of Assumptions

In summary, we use the following assumptions:A1The sea bottom is flat.A2A robot measures the relative position (bearing and range) of another robot using USBL sensors.A3The anchor robot can stand still while extending its legs into the sea bottom.A4While two robots fall to the sea bottom, they are connected using a cable, whose length is shorter than $r_{cs}$.A5A robot has active sonar sensors for detecting nearby obstacles.A6A robot has depth sensors for measuring its depth.

## 4. Cooperative Localization Based on the Stop–Go Policy

During the maneuver of a moving robot, the anchor robot stands still, while extending its legs into the sea bottom. In this way, the anchor robot can resist severe underwater currents. The moving robot localizes itself by measuring the relative position of the anchor robot.

The proposed stop–go policy iterates the following process. The moving robot sets a new waypoint and moves to reach the waypoint. While the moving robot travels, it calculates the relative position of the anchor robot using relative position measurements (USBL) and localizes itself. After the moving robot reaches the waypoint, the moving robot extends its legs into the sea bottom and is set as a new anchor robot. Then, the previous anchor robot becomes a new moving robot. The new moving robot approaches the anchor robot while localizing itself. Once the moving robot reaches the anchor robot, one iterates the process in this paragraph. Iterate this until the robots visit the goal point.

See Figure 1 for an illustration of the stop–go policy. In this figure, time elapses from top to bottom. In this figure, the movement of a moving robot is plotted with an arrow.

At the moment when a robot reaches the sea bottom, its position variance is initialized as $P_{0}$. The position variance of each robot is updated using (Equation 6). As time goes on, the variance of each robot increases using (Equation 6). This is inevitable since there exists sensor measurement noise in the system (Equation 6).

### 4.1. Guidance Laws for a Moving Robot, While Avoiding Obstacles

Before a moving robot begins maneuvering at sampling-stamp *k*, the robot needs to determine a new waypoint w(k) to visit. The waypoint w(k) is set, so that heading towards the waypoint can make the robot approach the goal while avoiding obstacles. Let $g\in R^{2}$ define the goal position defined in two dimensions (NE reference frame).

Once a waypoint is set, the robot begins moving toward the waypoint while overcoming underwater currents. Section 4.2 discusses how to make the robot move toward the waypoint.

To avoid collision with obstacles, the moving robot uses its active sonar sensors. The moving robot can use active sonar sensors to detect a point in obstacles. For instance, the moving robot can use active sonar sensors to detect the relative position of underwater nets [13]. Among all obstacle points, the moving robot detects the point, called the *closestPnt*, which is closest to the robot.

In the NE reference frame, let $C_{o}\left(k\right)\in R^{2}$ present the closestPnt at sampling-stamp *k*. Note that $C_{o}\left(k\right)$ can be measured utilizing the horizontal range sensors, such as active sonar sensors, with maximum sensing range $r_{s}$. Let $N\left(k\right)\in R^{2}$ be defined as $N\left(k\right)={\hat{r}}^{M}\left(k\right)-C_{o}\left(k\right)$.

Let $r_{0}$ define the separation threshold between the moving robot and an obstacle boundary (the boundary of an obstacle). Since an obstacle boundary is detected utilizing active sonar sensors of the moving robot, $r_{0}\leq r_{s}$ is required.

Suppose that the moving robot is sufficiently far from an obstacle. Then, the robot moves towards the goal. In other words, if $∥N\left(k\right)∥\geq r_{0}$, then the moving robot at sampling-stamp *k* sets the waypoint, $w\left(k\right)\in R^{2}$, as
(10)$$\begin{array}{c} w\left(k\right)={\hat{r}}^{M}\left(k\right)+\frac{g-{\hat{r}}^{M}\left(k\right)}{∥g-{\hat{r}}^{M}\left(k\right)∥}r_{cs}. \end{array}$$

Let $k_{o}$ define the sampling-stamp such that
(11)$$\begin{array}{c} ∥N\left(k_{o}-1\right)∥\geq r_{0} \end{array}$$
and that
(12)$$\begin{array}{c} ∥N\left(k_{o}\right)∥<r_{0}. \end{array}$$

At sampling-stamp $k_{o}$, the moving robot begins following the obstacle boundary.

Suppose that we want to make the moving robot follow the obstacle, while the obstacle is to the right of the robot. Then, the moving robot sets its waypoint w(k) as
(13)$$\begin{array}{c} w_{r}\left(k\right)={\hat{r}}^{M}\left(k\right)+R\left(\pi /2\right)\frac{N(k)}{∥N(k)∥}r_{cs}. \end{array}$$

Here,
(14)$$\begin{array}{c} R\left(\alpha \right)=\left(\begin{array}{cc} \cos(\alpha ) & \sin(\alpha )\\ -\sin(\alpha ) & \cos(\alpha ) \end{array}\right) \end{array}$$
is the matrix presenting α radians rotation. In (Equation 13), R(π/2) is multiplied so that the moving robot’s orientation vector is normal to N(k).

Suppose that we want to make the moving robot follow the obstacle, while the obstacle is to the left of the robot. Then, the moving robot sets its waypoint w(k) as
(15)$$\begin{array}{c} w_{l}\left(k\right)={\hat{r}}^{M}\left(k\right)+R\left(-\pi /2\right)\frac{N(k)}{∥N(k)∥}r_{cs}. \end{array}$$

Note that $w_{l}\left(k\right)-{\hat{r}}^{M}\left(k\right)$ in (Equation 15) is in the opposite direction from $w_{r}\left(k\right)-{\hat{r}}^{M}\left(k\right)$ in (Equation 13). We can set the waypoint w(k) considering the relative position of the goal with respect to the robot. In the case where
(16)$$\begin{array}{c} \left(w_{l}\left(k\right)-{\hat{r}}^{M}\left(k\right)\right)\cdot \left(g-{\hat{r}}^{M}\left(k\right)\right)>0 \end{array}$$
is met, then the moving robot sets its waypoint as $w\left(k\right)=w_{l}\left(k\right)$. Otherwise, the moving robot sets its waypoint as $w\left(k\right)=w_{r}\left(k\right)$.

It is desirable that the robot moves toward the goal if possible. Recall that $k_{o}$ is the sampling-stamp when the moving robot begins following the obstacle. While the moving robot follows the obstacle boundary under (Equation 13) or (Equation 15), it detects the moment when both
(17)$$\begin{array}{c} N\left(k\right)\cdot ({\hat{r}}^{M}\left(k\right)-g)<0 \end{array}$$
and
(18)$$\begin{array}{c} ∥{\hat{r}}^{M}\left(k\right)-g∥<∥{\hat{r}}^{M}\left(k_{o}\right)-g∥ \end{array}$$
are satisfied. This implies that the obstacle boundary, which the moving robot has been following, is opposite to the goal direction. At this moment, the robot begins moving toward the goal using (Equation 10), since the robot does not have to keep following the obstacle boundary.

As the line-of-sight between the robot and the goal point is not blocked by an obstacle, the waypoint is set as (Equation 10). In this manner, the robot can move towards the goal while avoiding collision with an obstacle.

If the waypoint w(k) is set by the moving robot under (Equation 10), (Equation 13), or (Equation 15), then the robot needs to move towards w(k). Due to underwater currents, it is not trivial to make the robot at sampling-stamp *k* move towards w(k). In other words, due to underwater currents, heading toward the waypoint does not necessarily make the robot move toward the waypoint. How to make the robot under underwater currents move towards the waypoint is presented in Section 4.2.

Figure 2 illustrates concepts used in the guidance controls. The triangle centered at ${\hat{r}}^{M}\left(k\right)$ shows the attitude of the moving robot. In Figure 2, $C_{o}\left(k\right)$ is depicted on the obstacle boundary. In addition, N(k) is plotted as an arrow emanating from $C_{o}\left(k\right)$. Suppose that the robot heads towards h in this figure. Due to underwater currents, the robot moves towards d instead of h.

### 4.2. Make the Robot under Underwater Currents Move towards the Waypoint

Due to underwater currents, heading toward the waypoint does not necessarily make the robot move toward the waypoint. In order to move towards the waypoint, the moving robot estimates the horizontal motion perturbation due to underwater currents.

While the robot moves, it localizes itself utilizing (Equation 5). Based on the localization result ${\hat{r}}^{M}$, the robot’s motion perturbation due to underwater currents is estimated in real-time.

One next presents how to estimate the horizontal motion perturbation due to underwater currents which apply to the moving robot. Suppose the robot moves with speed $v^{M}\left(k\right)$ and with orientation $\theta^{M}\left(k\right)$ at sampling-stamp *k*. Then, in the NE reference frame, the predicted position of the robot at sampling-stamp k+1 is
(19)$$\begin{array}{c} {\hat{r}}^{M}\left(k+1|k\right)={\hat{r}}^{M}\left(k\right)+T\times v^{M}\left(k\right)\mathrm{cs}\left(\theta^{M}\left(k\right)\right). \end{array}$$

Using (Equation 2), we get
(20)$$\begin{array}{c} r^{M}\left(k+1\right)=r^{M}\left(k\right)+T\times \left(f\left(r^{M}\left(k\right)\right)+v^{M}\left(k\right)\mathrm{cs}\left(\theta^{M}\left(k\right)\right)\right). \end{array}$$

However, accessing $r^{M}\left(k\right)$ or $r^{M}\left(k+1\right)$ is not feasible due to sensor noise. At each sampling-stamp, $r^{M}$ is estimated by measuring the relative position of the anchor robot. In other words, ${\hat{r}}^{M}$ is derived utilizing (Equation 5).

By replacing $r^{M}$ in (Equation 20) by ${\hat{r}}^{M}$, one gets
(21)$$\begin{array}{c} {\hat{r}}^{M}\left(k+1\right)={\hat{r}}^{M}\left(k\right)+T\times \left(f\left({\hat{r}}^{M}\left(k\right)\right)+v^{M}\left(k\right)\mathrm{cs}\left(\theta^{M}\left(k\right)\right)\right). \end{array}$$

Using (Equation 21) and (Equation 19), the horizontal motion perturbation due to underwater currents is estimated as
(22)$$\begin{array}{c} f\left({\hat{r}}^{M}\left(k\right)\right)=\frac{{\hat{r}}^{M}\left(k+1\right)-{\hat{r}}^{M}\left(k\right)}{T}-v^{M}\left(k\right)\mathrm{cs}\left(\theta^{M}\left(k\right)\right). \end{array}$$

Note that the accuracy of this perturbation estimation depends on the accuracy of location estimation since $r^{M}$ in (Equation 20) was replaced by ${\hat{r}}^{M}$.

Next, the variance of the perturbation estimation is derived. Using (Equation 22), one gets
(23)$$\begin{array}{c} V\left(f\left({\hat{r}}^{M}\left(k\right)\right)\right)=V\left(\frac{{\hat{r}}^{M}\left(k+1\right)-{\hat{r}}^{M}\left(k\right)}{T}\right)+V\left(v^{M}\left(k\right)\mathrm{cs}\left(\theta^{M}\left(k\right)\right)\right). \end{array}$$

One next addresses how to derive $V(\frac{{\hat{r}}^{M}\left(k+1\right)-{\hat{r}}^{M}\left(k\right)}{T})$ in (Equation 23). Using (Equation 5), we get
(24)$$V\left(\frac{{\hat{r}}^{M}\left(k+1\right)-{\hat{r}}^{M}\left(k\right)}{T}\right)=\frac{1}{T^{2}}\left(V\left(\begin{array}{c} r_{k+1}^{l}c\left(b_{k+1}^{l}\right)\\ r_{k+1}^{l}s\left(b_{k+1}^{l}\right) \end{array}\right)+V\left(\begin{array}{c} r_{k}^{l}c\left(b_{k}^{l}\right)\\ r_{k}^{l}s\left(b_{k}^{l}\right) \end{array}\right)\right).$$

Here, one used the fact that $V\left({\hat{r}}^{S}\left(k+1\right)\right)=V\left({\hat{r}}^{S}\left(k\right)\right)$, since the anchor robot does not move at all. Recall that (Equation 7) can be used to derive $V\left(\begin{array}{c} r_{k}^{l}c\left(b_{k}^{l}\right)\\ r_{k}^{l}s\left(b_{k}^{l}\right) \end{array}\right)$.

In underwater environments, the magnetic field may be distorted, which leads to inaccurate measurements of $\theta^{M}\left(k\right)$. In addition, there may be an error in the estimation of through-water speed $v^{M}\left(k\right)$. Therefore, $V(v^{M}\left(k\right)\mathrm{cs}\left(\theta^{M}\left(k\right)\right))$ in (Equation 23) is not zero. $V(v^{M}\left(k\right)\mathrm{cs}\left(\theta^{M}\left(k\right)\right))$ in (Equation 23) can be estimated utilizing extensive experiments.

Once the horizontal motion perturbation due to underwater currents is estimated utilizing (Equation 22), then the estimated perturbation is further used to make the robot at sampling-stamp *k* move towards the waypoint w(k). Recall that at each sampling-stamp *k*, the waypoint w(k) is set using (Equation 10), (Equation 13), or (Equation 15). Since one desires that the robot moves towards the waypoint w(k), the desired position of the robot at sampling-stamp k+1 is
(25)$$\begin{array}{c} d={\hat{r}}^{M}\left(k\right)+T\times v^{M}\left(k\right)\mathrm{cs}\left(\theta^{d}\left(k\right)\right). \end{array}$$

Here, $\mathrm{cs}\left(\theta^{d}\left(k\right)\right)=\left(\begin{array}{c} c(\theta^{d}\left(k\right))\\ s(\theta^{d}\left(k\right)) \end{array}\right)$ where
(26)$$\begin{array}{c} \theta^{d}\left(k\right)=\tan^{-1}\left(\left(w\left(k\right)-{\hat{r}}^{M}\left(k\right)\right)\left[2\right],\left(w\left(k\right)-{\hat{r}}^{M}\left(k\right)\right)\left[1\right]\right). \end{array}$$

Figure 2 illustrates the concepts. In Figure 2, the waypoint w(k) is depicted as a circle. The robot sets its heading point as
(27)$$\begin{array}{c} h=d-f({\hat{r}}^{M}\left(k\right))\times T. \end{array}$$

In Figure 2, the triangle centered at ${\hat{r}}^{M}\left(k\right)$ shows the attitude of the robot in the case where the robot sets its heading point as h. However, due to underwater currents, the robot moves towards $d=h+f({\hat{r}}^{M}\left(k\right))\times T$ instead of h. This implies that the robot moves towards w(k).

As depicted on Figure 2, the orientation direction of the moving robot is set as
(28)$$\begin{array}{c} \theta^{M}\left(k\right)=\tan^{-1}\left(\left(h-{\hat{r}}^{M}\left(k\right)\right)\left[2\right],\left(h-{\hat{r}}^{M}\left(k\right)\right)\left[1\right]\right). \end{array}$$

Moreover, the robot’s speed $v^{M}\left(k\right)$ is changed to $∥(h-{\hat{r}}^{M}\left(k\right))∥/T$. In this manner, the robot can move towards w(k) considering the motion perturbation due to underwater currents.

## 5. Simulations

This section verifies the effectiveness of the proposed approach utilizing MATLAB simulations. Since we consider a flat sea bottom, one considers localization of a robot in the NE reference frame. Initially, one robot is at (35,35), and the other robot is at (40,40) meters. $r_{cs}=30$ m, and $r_{0}=50$ m. In addition, one sets T=10 s.

We verify the robustness of the proposed strategy by setting noise parameters as follows: rngS=0.1 m, and brgS=0.5 degrees. These sensor specifications are available using USBL sensors.

The initial covariance matrix is set as $P_{0}=\mathrm{diag}\left(100,100\right)$. The simulation ends when a robot reaches the goal g.

Recall that $f(r)\in R^{2}$ in (Equation 20) defines the perturbation on the robots’ position $r\in R^{2}$ due to underwater currents. It is not trivial to simulate the complex interaction between the underwater currents and the motion of a hexapod robot on the sea bottom. In addition, it is not trivial to simulate the complex interaction between underwater currents and obstacles, such as underwater nets. We simulate the perturbation vector field f(r) in a rectangular 2D workspace with size 1200 × 1200 m.

This section simulates scenarios where the underwater currents change irregularly. One discusses how to generate the irregular perturbation vector field. Let s=1200 (m) denote the side length of the rectangular workspace. Since $r\in R^{2}$ exists inside the rectangular workspace, r[1] and r[2] exist in the interval [0,s]. Let $flow_{max}=0.1$ (m/s) denote the maximum flow speed. The function f(r) is calculated using the following equations.
(29)$$\begin{array}{ccc} f(r)[1] & = & noiseS*randn+flow_{max}\frac{\sqrt{r{\left[1\right]}^{2}+r{\left[2\right]}^{2}}}{s\sqrt{2}}. \end{array}$$
(30)$$\begin{array}{ccc} f(r)[2] & = & noiseS*randn+flow_{max}\frac{s\sqrt{2}-\sqrt{r{\left[1\right]}^{2}+r{\left[2\right]}^{2}}}{s\sqrt{2}}. \end{array}$$

Here, noiseS indicates the noise strength in the flow field. In addition, randn indicates a Gaussian noise with zero mean and variance 1. We use noiseS=0.01 in simulations. Due to the noise term noiseS*randn, a distinct perturbation vector field is generated whenever one runs a scenario simulation. Blue arrows in the following figures (Figures 3, 4, 7, 10 and 13) show the perturbation vector field that is irregularly generated in the environment.

Note that the maximum perturbation speed must be slower than the speed of a moving robot. Otherwise, it is impossible to make a moving robot overcome underwater currents. (Equation 2) is used to simulate the motion of a robot under underwater currents. In MATLAB simulations, the speed of a moving robot is $v_{i}\left(k\right)=0.3$ m/s, which is faster than the maximum perturbation speed.

Figure 3 depicts the overview of the simulation environment. The origin of the rectangular workspace is located at the left bottom corner of the workspace. Blue arrows show the perturbation vector field in the environment. In addition, obstacle boundaries are depicted in red.

### 5.1. Scenario 1

This subsection simulates the case where the goal is located at (80,80) meters. Figure 4 presents the simulation result in the environment of Figure 3. Figure 4 is the enlarged figure of Figure 3. In Figure 4, the path of one robot is depicted with blue asterisks, and that of another robot is depicted with green asterisks. See that both robots move towards the goal while overcoming underwater currents. The path of one robot overlaps with that of another robot since both robots visit identical waypoints sequentially. Red dotted arrows in the figure show the perturbation vector field estimated at the anchor robot’s position. It is acknowledged that the perturbation direction is not estimated accurately. This is due to the fact that the true robot position is not available due to sensor noise in the system. Equation (Equation 23) presents the variance in the perturbation velocity estimation.

Figure 5 presents the localization error $∥r_{i}\left(k\right)-{\hat{r}}_{i}\left(k\right)∥$ with respect to sampling-stamp *k*. Here, ${\hat{r}}_{i}\left(k\right)$ is an estimate of the true position $r_{i}\left(k\right)$. Under underwater currents, dead reckoning localization based on inertial navigation sensors results in the accumulation of localization errors. Figure 5 shows that the error accumulation under the stop–go policy is not severe. Note that sensor noise specifications (rngS=0.1 m, and brgS=0.5 degrees) are used, which are available using USBL sensors.

The orientation direction of the *i*-th robot at sampling-stamp *k* is $\mathrm{cs}(\theta_{i}\left(k\right))$ in (Equation 2). Let $head_{x}$ represent $c(\theta_{i}\left(k\right))$, and let $head_{y}$ represent $s(\theta_{i}\left(k\right))$. Figure 6 plots both $head_{x}$ and $head_{y}$ of each robot at each sampling-stamp *k*. While a robot is settled down, its $head_{x}$ and $head_{y}$ are set as zeros. See that the orientation of a moving robot changes according to the estimated underwater currents.

#### Comparison with Dead Reckoning Localization Methods

For comparison, we run Scenario 1 using dead reckoning localization methods. For all i∈{1,2}, $\mathrm{cs}(\theta_{i}\left(k\right))$ in (Equation 2) is set as $\frac{g-{\hat{r}}_{i}\left(k\right)}{∥g-{\hat{r}}_{i}\left(k\right)∥}$. Note that (Equation 2) is used to simulate the motion of a robot under underwater currents. In MATLAB simulations, the speed of a moving robot is $v_{i}\left(k\right)=0.3$ m/s. A robot cannot move towards the goal, due to the perturbation of underwater currents. See $f(r_{i}\left(k\right))$ term in (Equation 2).

Figure 7 presents the simulation result in the environment of Figure 3. Figure 7 is the enlarged figure of Figure 3. In Figure 7, the path of one robot is depicted with blue asterisks, and that of another robot is depicted with green asterisks. See that both robots cannot reach the goal, which is marked with a black asterisk. Therefore, the simulations end when 700 s (70 sampling-stamps) elapse.

Figure 8 depicts the localization error $∥r_{i}\left(k\right)-{\hat{r}}_{i}\left(k\right)∥$ with respect to sampling-stamp *k*. Under underwater currents, dead reckoning localization based on inertial navigation sensors results in a severe accumulation of localization errors. Figure 9 plots both $head_{x}$ and $head_{y}$ with respect to sampling-stamps. See that the orientation of a moving robot does not change as time goes on. Since the perturbation direction is not considered, a robot does not head toward the goal.

### 5.2. Scenario 2

We verify the robustness of the proposed strategy in complex environments with various obstacles. We simulate the case where the goal is located at (1100,900) meters. Figure 10 presents the simulation result. The path of one robot overlaps with that of another robot since both robots visit identical waypoints sequentially. See that two robots approach the goal while avoiding obstacles. Red dotted arrows in the figure show the perturbation vector field estimated at the anchor robot’s position. It is acknowledged that the perturbation direction is not estimated accurately, since the true robot position is not available due to sensor noise in the system. Note that sensor noise specifications (rngS=0.1 m, and brgS=0.5 degrees) are used, which are available using USBL sensors.

Figure 11 presents the localization error $∥r_{i}\left(k\right)-{\hat{r}}_{i}\left(k\right)∥$ with respect to sampling-stamp *k*. See that the localization error remains a small value at all sampling-stamps. Figure 12 plots both $head_{x}$ and $head_{y}$ with respect to sampling-stamps. While a robot is settled down, its $head_{x}$ and $head_{y}$ are set as zeros. The orientation of a moving robot changes according to the estimated perturbation due to underwater currents.

### 5.3. Scenario 3

We verify the robustness of the proposed strategy in complex environments with large perturbation noise. Recall that the perturbation vector field was generated by adding Gaussian noise with zero mean and standard deviation noiseS=0.01 to the noiseless perturbation vector. In Scenario 3, the perturbation vector field was generated by adding Gaussian noise with zero mean and standard deviation noiseS=0.05 to the noiseless perturbation vector.

We simulate the case where the goal is located at (1100,900) meters. Figure 13 depicts the simulation result. The path of one robot overlaps with that of another robot since both robots visit identical waypoints sequentially. See that two robots approach the goal while avoiding obstacles. It is acknowledged that the perturbation direction is not estimated accurately, since the true robot position is not accessible due to sensor noise in the system.

Figure 14 presents the localization error $∥r_{i}\left(k\right)-{\hat{r}}_{i}\left(k\right)∥$ with respect to sampling-stamp *k*. See that the localization error remains a small value at all sampling-stamps. The localization error in Scenario 3 is comparable to that in Scenario 2 (Figure 11), which verifies the robustness of the proposed location scheme.

Figure 15 plots both $head_{x}$ and $head_{y}$ with respect to sampling-stamps. While a robot is settled down, its $head_{x}$ and $head_{y}$ are set as zeros. The orientation of a moving robot changes according to the estimated perturbation due to underwater currents.

## 6. Conclusions

This paper considers underwater environments where there are no stable landmarks or features which can assist a robot’s localization. Furthermore, due to underwater currents, dead reckoning localization based on inertial navigation sensors results in a severe accumulation of localization errors.

This paper proposes novel cooperative localization schemes of two underwater hexapod robots together with a guidance strategy for avoiding obstacles under underwater currents. As for cooperation localization, the stop–go policy is developed to avoid the accumulation of localization errors due to the conventional dead reckoning approach based on inertial navigation sensors.

The estimation of motion perturbation due to underwater currents is taken into account while developing the guidance strategy for avoiding obstacles. The effectiveness of the proposed stop–go policy and guidance strategy is verified using MATLAB simulations. MATLAB simulations demonstrate that the proposed schemes are effective in that the current magnitude changes irregularly.

Since the proposed algorithms are simple, the computational load of our algorithms is very low. In addition, the algorithms can be applied for simultaneous localization and guidance for aerial vehicles or ground vehicles in environments with no landmarks or features. In the future, the effectiveness of the proposed methods will be verified using underwater experiments with real hexapod robots.

## Figures and Tables

**Figure 1 sensors-23-03186-f001:**
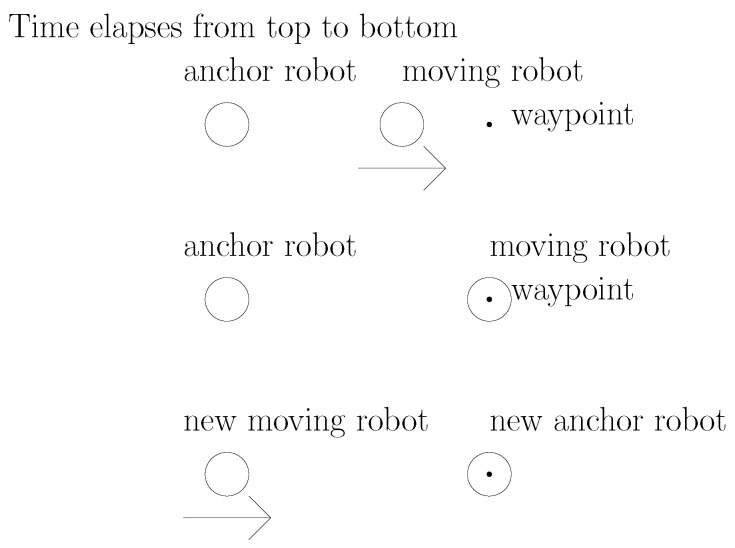
An illustration of the stop–go policy. Time elapses from top to bottom. The movement of a moving robot is plotted with an arrow.

**Figure 2 sensors-23-03186-f002:**
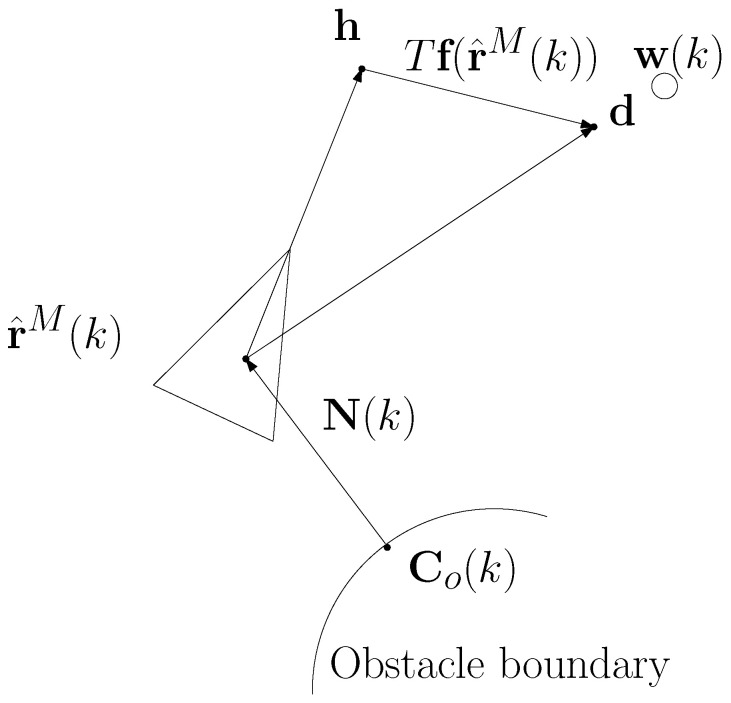
An illustration of concepts used in the guidance controls. The triangle centered at ${\hat{r}}^{M}\left(k\right)$ shows the attitude of the moving robot. In this figure, $C_{o}\left(k\right)$ is depicted on the obstacle boundary. In addition, N(k) is plotted as an arrow emanating from $C_{o}\left(k\right)$. Suppose that the robot heads towards h in this figure. Due to underwater currents, the robot moves towards d instead of h. This implies that the robot moves towards the waypoint w(k), which is depicted as a circle.

**Figure 3 sensors-23-03186-f003:**
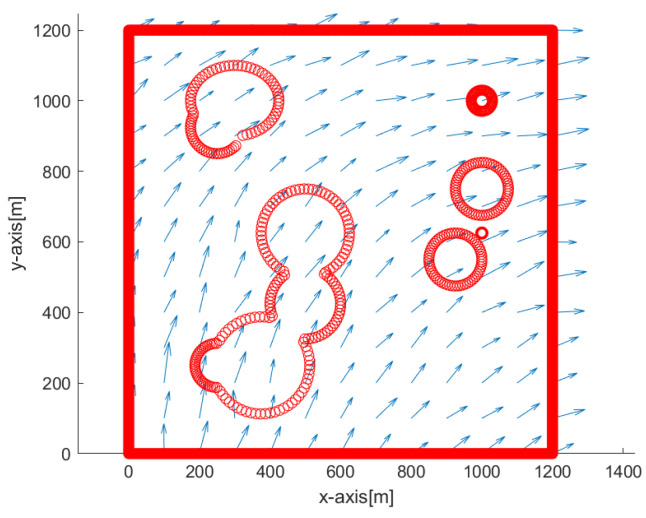
The origin of the rectangular workspace is located at the left bottom corner of the workspace. Blue arrows show the perturbation vector field that is irregularly generated in the environment. Obstacle boundaries are depicted with red circles.

**Figure 4 sensors-23-03186-f004:**
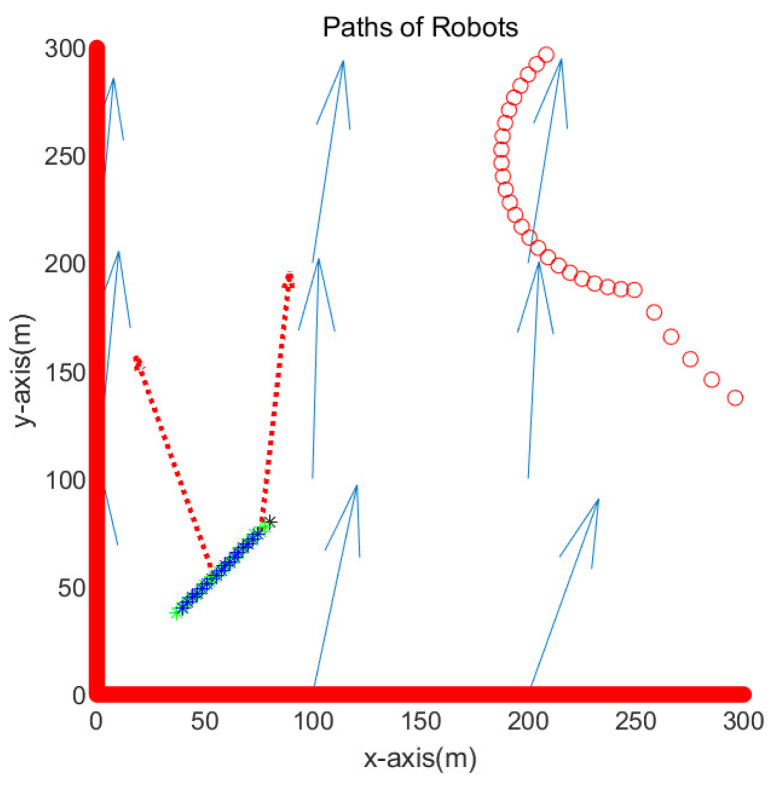
Scenario 1. The path of one robot is depicted with blue asterisks, and that of another robot is depicted with green asterisks. Red dotted arrows show the perturbation vector field estimated at the robot’s position.

**Figure 5 sensors-23-03186-f005:**
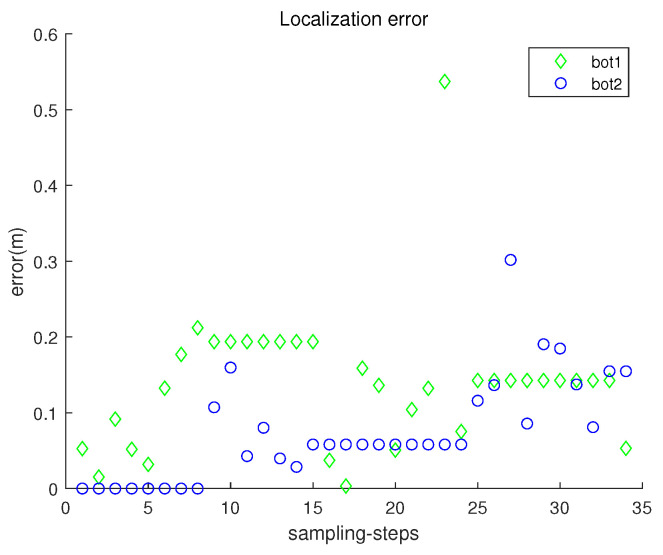
Scenario 1. The localization error $∥r_{i}\left(k\right)-{\hat{r}}_{i}\left(k\right)∥$ with respect to sampling-stamp *k*.

**Figure 6 sensors-23-03186-f006:**
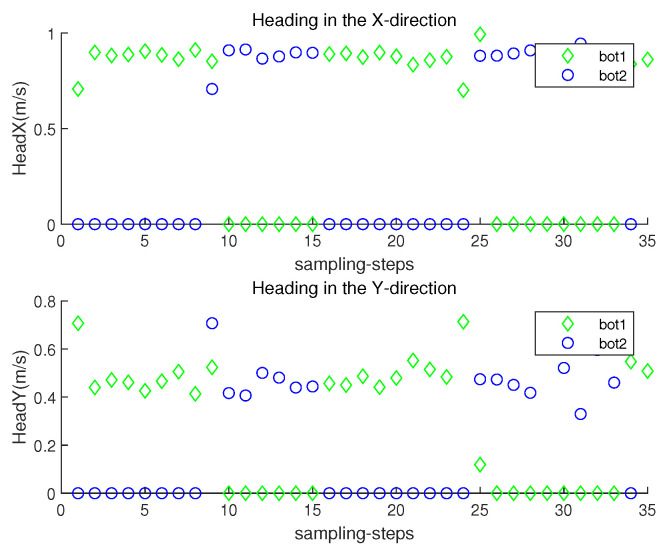
Scenario 1. $head_{x}$ and $head_{y}$ with respect to sampling-stamps.

**Figure 7 sensors-23-03186-f007:**
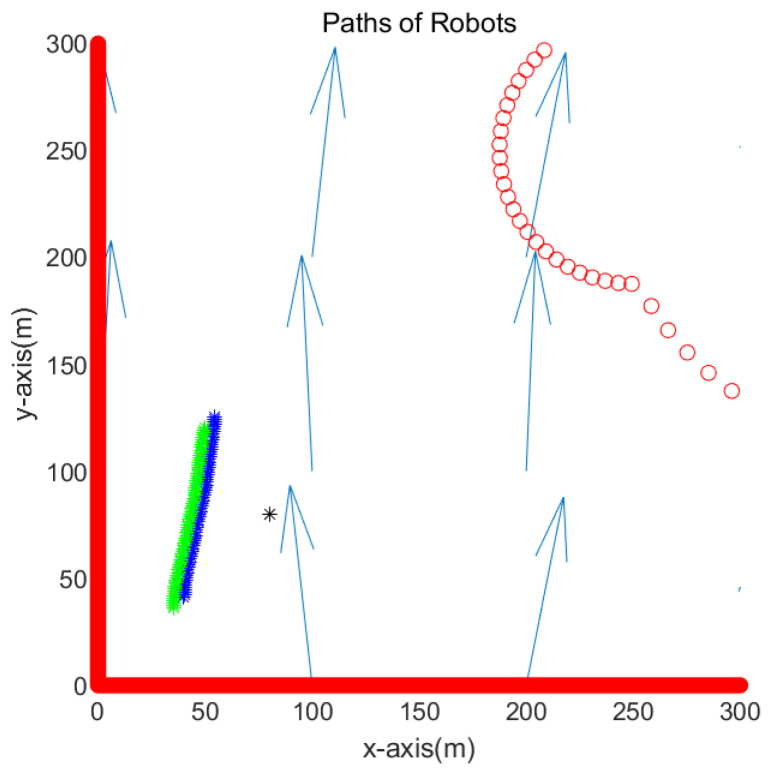
Scenario 1. Dead reckoning localization methods are used. The path of one robot is depicted with blue asterisks, and that of another robot is depicted with green asterisks. The goal is marked with a black asterisk.

**Figure 8 sensors-23-03186-f008:**
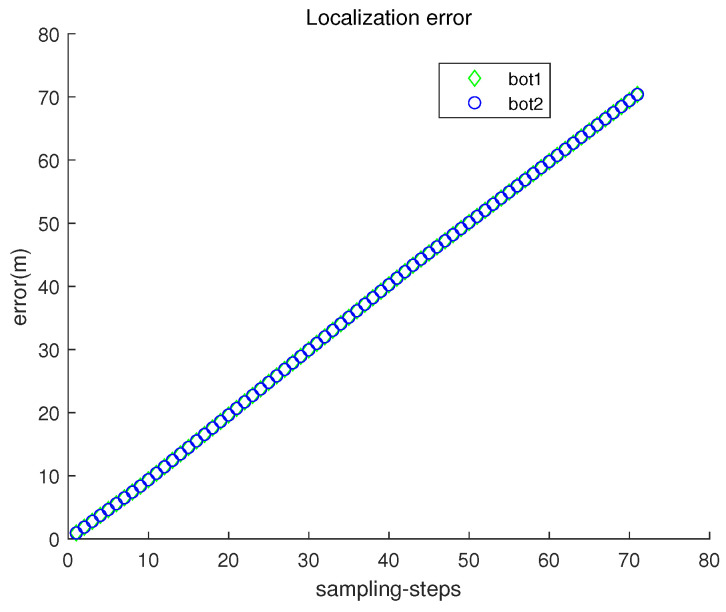
The localization error $∥r_{i}\left(k\right)-{\hat{r}}_{i}\left(k\right)∥$ with respect to sampling-stamp *k* (dead reckoning strategy in Scenario 1).

**Figure 9 sensors-23-03186-f009:**
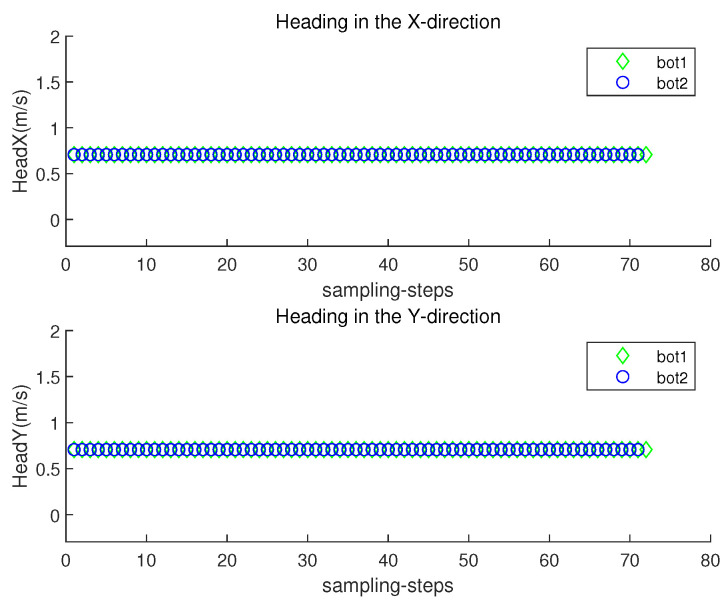
$head_{x}$ and $head_{y}$ with respect to sampling-stamps (dead reckoning strategy in Scenario 1).

**Figure 10 sensors-23-03186-f010:**
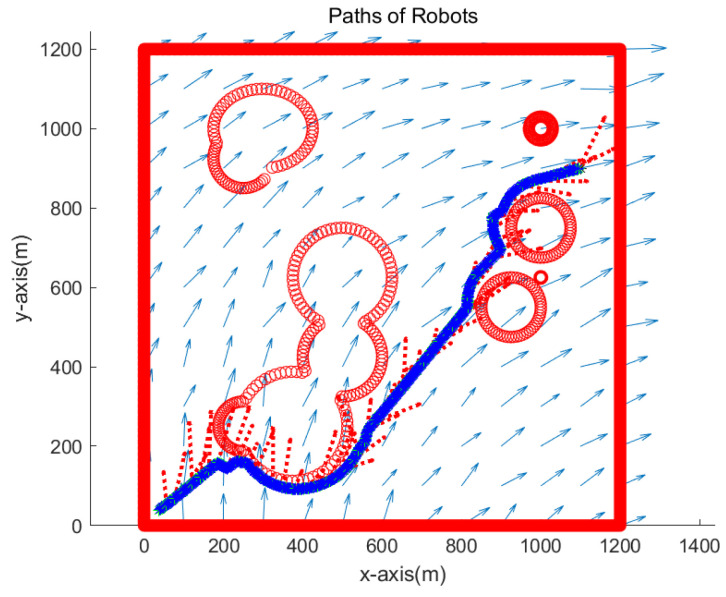
Scenario 2. The path of one robot is depicted with blue asterisks, and that of another robot is depicted with green asterisks. Red dotted arrows show the perturbation vector field estimated at the anchor robot’s position.

**Figure 11 sensors-23-03186-f011:**
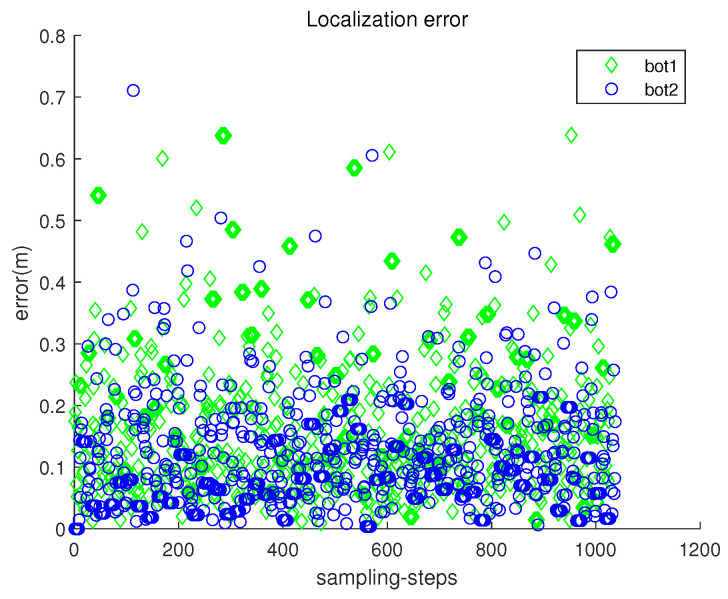
Scenario 2. The localization error $∥r_{i}\left(k\right)-{\hat{r}}_{i}\left(k\right)∥$ with respect to sampling-stamp *k*.

**Figure 12 sensors-23-03186-f012:**
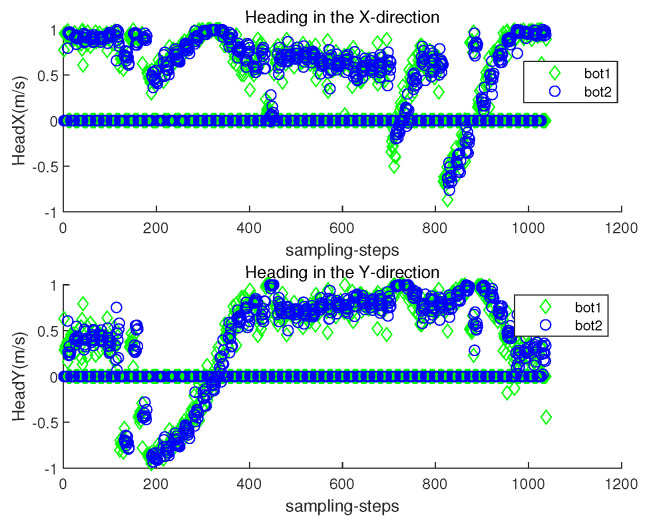
Scenario 2. $head_{x}$ and $head_{y}$ with respect to sampling-stamps.

**Figure 13 sensors-23-03186-f013:**
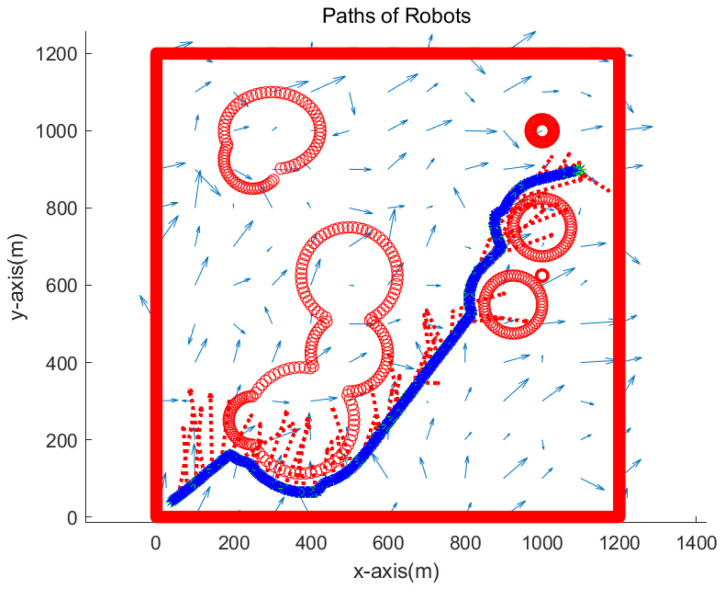
Scenario 3. In Scenario 3, the perturbation vector field was generated by adding Gaussian noise with zero mean and standard deviation of 0.05 to the noiseless perturbation vector. The path of one robot is depicted with blue asterisks, and that of another robot is depicted with green asterisks. See that two robots approach the goal while avoiding obstacles.

**Figure 14 sensors-23-03186-f014:**
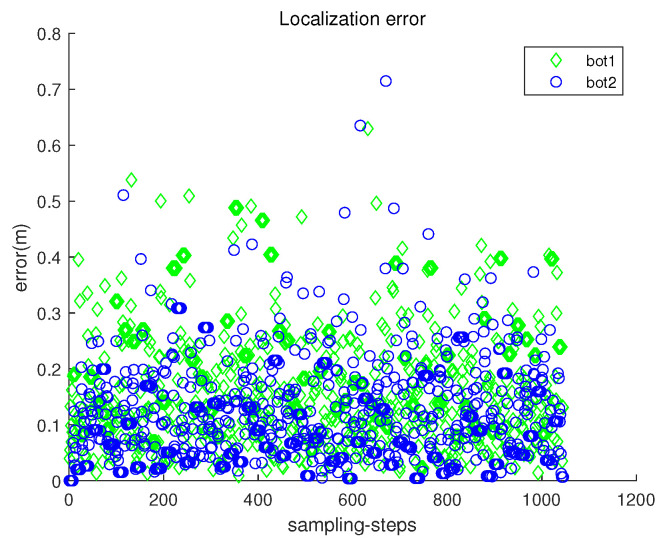
Scenario 3. The localization error $∥r_{i}\left(k\right)-{\hat{r}}_{i}\left(k\right)∥$ with respect to sampling-stamp *k*. The localization error in Scenario 3 is comparable to that in Scenario 2 (Figure 11), which verifies the robustness of the proposed location scheme.

**Figure 15 sensors-23-03186-f015:**
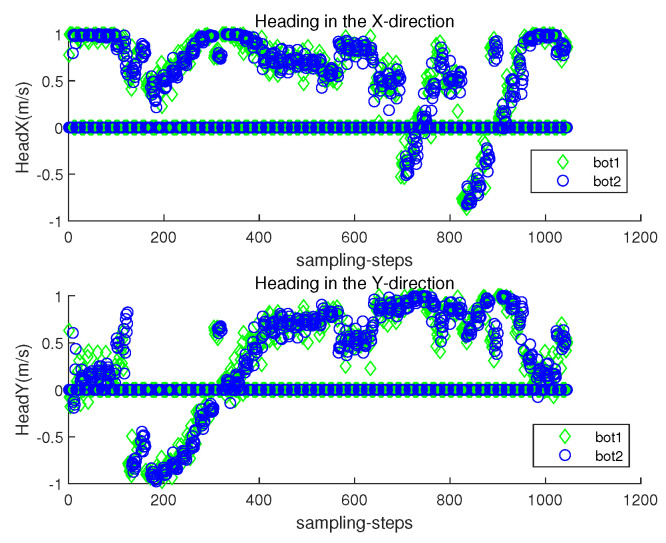
Scenario 3. $head_{x}$ and $head_{y}$ with respect to sampling-stamps.

## Data Availability

Not applicable.

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
