# Peer review of "Simultaneous Localization and Guidance of Two Underwater Hexapod Robots under Underwater Currents"

_sensors, 2023, doi:10.3390/s23063186_

Round 1
Reviewer 1 Report
This manuscript addresses the simultaneous localization and guidance of two underwater hexapod robots under sea currents by considering an underwater environment where there are no landmarks or features which can assist a robot’s localization. The manuscript is with novelty and well organized, which can be accepted by the following revisions:
1 Boundary conditions need to be clearly described.
2 The light green lines and data points in Figures 4, 5, 10, 11, 13, and 14 cannot be seen clearly.
3 How to choose the optimal number of sensors?
4 The definitions of error, variance and uncertainty need to be explained.
5 Recent research progress on optimal sensor placement should be referred: 10.1016/j.ymssp.2019.01.057 and 10.1016/j.ymssp.2021.108386.
6 Articles (a, an, the) are missing in many places in the manuscript and need to be written accurately.
Author Response
Thank you very much for your valuable comments. The response to Reviewer 1 is attached.

Reviewer 2 Report
The topic addressed by the article is quite interesting and relevant. The proposed methods, at first glance, do seem to represent an innovative contribution. However, the work is not well presented and there are flaws in the organization. Some important parts were not inserted.
At the beginning of Section 2, the authors make some important assumptions but fail to make the fundamental assumption that robots can measure their positions using sonar sensors. This hypothesis, which supports the measurement and estimation methods, only appears at the beginning of Section 2.2.
The results presented in the simulation section are somewhat superficial, and there are some missing detail about the robot dimensions, about the coordinated systems, about the region in which they interact. The way the text is presented, it would be quite difficult to reproduce the simulation results by other collaborators or readers interested in the topic.
Comments:
Page 4, Section 2: The text is a little confusing. It is not clear what the definitions are and what the assumptions are. I suggest that authors rearrange the definitions and assumptions using a suitable formatting environment. For example, "\begin{definition} … \end{definition}" and "\begin{assumption} … \end{assumption}". This is important for enumeration and facilitates citations of them.
Page 4, line 157: I suggest authors no to begin a sentence with a math function. "The function $tan^{-1}(y,x)$ ..." would be better. Please, consider the same comment for the second and third sentences within lines 160-163.
Page 4, line 160: Its is very unusual to define a diagonal matrix as "D(a, b, c., , , )". I suggest the authors to adopt the notation "$D=diag \{a_1,a_2,\cdots,a_n\}$ in which $a_1,\cdots,a_n$ appear as its diagonal elements"
Page 4, Section 2.1: Some notations are a little complicated. For instance the notation "[1:2]", adopted in Eq.(1) and others, could be omitted. Since 2 robots are used, the authors could simplify the presentations by writing the two equations, one for each robot.
In sections 2.1 and 2.2 there are some crucial points that the authors ignored in the presentation. First, an illustrative image of the system is missing. This would clarify and highlight important variables such as distances and dimensions of the robots. But the main issue is that the coordinated systems of each robot have not been mentioned or shown. This is very important to define distances, the Threat-zone and other variables of interest. I strongly recommend authors to clarify such issues. The description of the proposed methods is difficult to follow and understand without the figures and diagrams illustrating the system.
Page 5, lines 218-219: Again, r^s(k) and r^M(k) are meaningless without a reference coordinated system.
Page 5, Eq.(3): The notations "[1],[2]" presented in the tan^{-1} formula seem to refer to the components "1" and "2" of the vector r^{rel}(k). This form of notation is not usual and needs to be defined a priori in the text. Please, consider the same comment for Eqs.(5),(6).
Page 9, Figure 1: Caption has repeted sentences.
Page 10, line 379. Please, avoid starting sentences with equation numbers.
Page 11, Section 4: Simulations are not well described in this section. Simuation environment needs to be detached before presenting the results. For instance, authors may orientate themselves by answering the following issues:
- What are the dimensions of the sea terrain considered in the simulation tests ? Is it rectangular, circular or has it another geometric profile ?
- In which point of the terrain authors have fixed the reference coordinated system ? It seems that was (0,0)! But is this at the center or at a corner ?
- What are the physical dimensions of the robots ? Are they considered as particles in the simulation tests ?
Author Response
Thank you very much for your valuable comments. The response to Reviewer 3 is attached.

Reviewer 3 Report
- Please rewrite the abstract by reducing the background information and enhancing the details on your own work. Also, add a couple of sentences at the end of the abstract summarizing the results achieved.
- Please include recent state-of-the-art on SLAM for underwater environments e.g. DOI: 10.1109/ICRA46639.2022.9811695 and 10.1016/j.cosrev.2022.100510. Also, in Section 0 (Introduction), Line 30, SLAM may be introduced with reference to https://doi.org/10.5755/j01.eee.20.9.8707
- The discusison on results needs to be more critical and must be mentioned in quantitative manner.
- Readability of many figures (e.g. 4, 5, 7, 8, 10, 14) needs to be enhanced.
- Please let the paper proofread by a professional or a native writer for linguistic improvements.
Author Response

(The authors gave the same response as above.)

Round 2
Reviewer 2 Report
This new version of the manuscript is better organized. Most of the issues raised and revisions suggested by this reviewer were satisfactorily addressed. However, I note that there is one important point that the authors did not include: details about the coordinate system. At Page 11, line 414, the authors mention: "... is the NE reference frame ...". This information is vague. The coordinate system (fixed or moving) must be fixed at some point in space. This should reveal, for example, the directions of the X,Y,Z coordinate axes and in which directions vectors are defined.
I suggest that the authors define this system using the "NE" direction as a reference. Choose a fixed or moving point and set the X, Y, Z directions conveniently. This will greatly enrich the presentation of the final version of the text.
Author Response
Thank you very much for your valuable comments. We consider two hexapod robots that move on a flat sea bottom. Hence, this paper considers the localization problem of a robot in 2D environments. We use an inertial reference frame whose origin is an arbitrary point on the flat sea bottom. The frame consists of two coordinates: one represents the position along the northern axis, another one along the eastern axis. This frame is called the North-East (NE) reference frame.